# Exploring Change in Children’s and Art Therapists’ Behavior during ‘Images of Self’, an Art Therapy Program for Children Diagnosed with Autism Spectrum Disorders: A Repeated Case Study Design

**DOI:** 10.3390/children9071036

**Published:** 2022-07-12

**Authors:** Celine Schweizer, Erik J. Knorth, Tom A. Van Yperen, Marinus Spreen

**Affiliations:** 1Research Centre for Small n-Designs, NHL-Stenden University of Applied Sciences, 8917 DD Leeuwarden, The Netherlands; marinus.spreen@nhlstenden.com; 2Faculty of Behavioral and Social Sciences, Groningen University, 9712 CP Groningen, The Netherlands; ej@knorth.nl (E.J.K.); t.vanyperen@nji.nl (T.A.V.Y.)

**Keywords:** art therapy, children, autism spectrum disorders, change processes, OAT, EAT

## Abstract

(1) Background: ‘Images of Self’ (IOS) is a recently developed and evaluated art therapy program of 15 sessions to reduce difficulties in ‘sense of self’, ‘emotion regulation’, ‘flexibility’, and ‘social behavior’ of children diagnosed with Autism Spectrum Disorders (ASD). In this paper, it is explored whether change in the child’s behaviors corresponds to the therapist’s actions during IOS and 15 weeks later. (2) Method: In a repeated case study design, twelve children and seven therapists participated. Art therapists monitored their own and the children’s behavior by applying two observation instruments: the OAT (Observation of a child with autism in Art Therapy) and EAT (Evaluation of Art Therapist’s behavior when working with a child with autism). Child behaviors during art making were—individually and as a group—compared with therapist’s actions at three moments during the program. (3) Results: Ten of twelve children showed a substantial or moderate positive behavior change considering all OAT subscales at the end of the program and 15 weeks after treatment. Improvement of ‘social behavior’ stood out. Halfway treatment art therapists most prominently showed support of ‘emotion regulation’, ‘flexibility’, and ‘social behavior’. Clear one-on-one relationships between changes in children’s behavior and actions of therapists could not be identified. (4) Conclusion: The study provides new insights in the AT treatment process by monitoring children’s and therapists’ behavior. The art making itself and the art therapy triangle (child, art making, therapist) offer opportunities to improve verbal and nonverbal communication skills of the child.

## 1. Introduction

Children with autism-related problems are often referred to art therapy (AT) in the Netherlands [1]. Art therapies are recommended in the Dutch Guidelines for Mental Health [2]. The American Art Therapy Association [3] states that people with autism are an important target group for (research into) AT.

In the DSM-5 [4], persons with Autism Spectrum Disorders (ASD) frequently have social-communicative deficits and repetitive/restricted behaviors and interests.

Recently, several studies have indicated that AT can provide a successful treatment for children with ASD-related problems. In a systematic review [5], two single group studies reported improvement of the children to engage in social situations and improve ability to focus attention. The review also included an RCT that showed some favorable but not significant outcomes for the treatment group, compared to the control group. The lack in the RCT is not unexpected, because of the variety of ASD features in children [6,7,8,9], which makes it hard to form a comparable treatment and control group. An extensive practice- and theory-based study evaluated the contribution of sensory experiences when touching art materials. This resulted in improvement of communication with the therapist [10]. Art making in AT supports the pleasure of the child and contributes to an improved therapeutic relationship. It also increases the range of patterns in the child’s behavior and art expressions [11,12].

Until recently, there was a gap in the knowledge regarding behavioral changes of children with ASD in AT, the activities of the art therapist, and the relationship between the two [1,13,14]. To address this challenge, a series of studies was performed, resulting in the AT program ‘Images of Self’ (IOS). This title refers to the important role of art making in the triangular relationship between client, art work, and art therapist. Every art making process and every art product mirrors the experiences of the maker within this triangular relationship.

IOS was developed firstly with two studies examining knowledge from experienced art therapists as well as literature about AT practices with children diagnosed with ASD [15,16]. Based on these studies, the building blocks of the IOS program were further articulated as consensus-based typical elements of AT [17]. IOS consists of 15 individual sessions with an art therapist who has been intensively trained to apply the program. Characteristics of the treatment and the relevant actions of the art therapist are described in a manual [18]. The starting point of the treatment is attunement to the needs and art expressions of the child.

The IOS program offers a frame that allows adjustments to individual needs, because every child with autism has different interests, skills and varied reasons for being referred to art therapy. For instance, an 11-year-old girl and her environment suffered from her emotional outbursts. An important step during her IOS treatment was to make a colorful felt blanket that she wanted to use to comfort herself. During the creation of this blanket, she moved the soft wool with soap and water until it turned into felt. She enjoyed the creation of the blanket. During this process, she talked with the art therapist and became more aware of what gives her (emotional) stress and what brings relaxation. The girl became aware that sensory experiences helped her to relax. She also made a small plastic bag with smooth glue in it, for keeping in her pocket. Whenever she became aware that her tension grew, it helped her to relax by touching the smooth bag.

The IOS program yielded promising results in a multiple case study among 12 children diagnosed with ASD, aged 6–12 years, with normal/high intelligence profiles [19]. Children were referred to IOS because of difficulties with their ‘sense of self’ (difficulties with reflecting on their own feelings and behaviors), ‘self-esteem’ (strongly negative senses about ‘being different’ but not understanding why), ‘emotion regulation problems’ (being very depressed; emotional outbursts), ‘flexibility problems’ (being upset when something unexpected happens) and/or ‘social problems’ (difficulties in expressing themselves and troubles with understanding others). Nearly all participating children (*n* = 11) started the program with severe problems in these areas according to the norms of the Child Social Behavior Questionnaire (CSBQ) [20].

During the program, the children were monitored on the outcomes ‘sense of self’, ‘emotion regulation’, ‘flexibility’, and ‘social behavior’ by therapists, but also by their parents and teachers [18]. The outcome ‘sense of self’ is a concept that includes a continuum regarding self-development: self-perception, self-image, self-concept, and self-esteem [21].

Seven of the 12 children significantly improved in ‘flexibility’ and ‘social behavior’, both during treatment and also 15 weeks after termination of the treatment [19]. These results can be interpreted as positive, because in general, the reduction rate of problems by psychosocial interventions in children and youth lies between 35 and 62 percent a year after the start of a treatment [22,23]. According to the qualitative analysis of the evaluation of IOS, all children were reported by their parents, teachers, and therapists to be happier and more stable, and better able to give words to their experiences. Additionally, improvements in ‘emotion regulation’ (*n* = 8) and ‘flexibility’ (*n* = 4) were reported.

In our evaluation study, the focus was primarily on the outcomes: do the children improve or not? However, AT is characterized by a triangular relationship between client, art, and therapist [24,25]. In this relationship, it is supposed that communication of the children with the art therapist will be easier and feel safer for the child, because of the component of art making. This especially applies to children with communication challenges such as children diagnosed with ASD [17]. In this paper, we explore the process of therapeutic change, thereby directing the attention to the development of children’s and therapists’ behavior in relation to each other. The research question in the current study is: *To what extent are changes in the behaviors of the children and the art therapists’ activities concerning (supporting) ‘sense of self’, ‘emotion regulation’, ‘flexibility’ related?*

## 2. Methods

### 2.1. Participants

The repeated case study design makes it possible to monitor individual patterns of results in different contexts. In our design, we included children, their parents, teachers and their art therapists. The multiple perspectives and the pre-test/post-test provide insight in changes in AT as well as in the contexts of home and school. Twelve children and seven therapists participated in this study. Children aged 6–12 years with an ASD diagnosis and an IQ ≥ 80 were included in the sample [19] (Table A1 in Appendix A). Children were excluded if they were evaluated by their art therapists as showing amounts of resistance that were too high or fear of art making. All therapists had a Bachelor’s degree in art therapy, which is the required professional qualification in the Netherlands for working as an art therapist. They also had at least two years of work experience in AT with the target group. Parents and teachers of the participating children completed questionnaires, observed daily behavior, and reported possible behavior changes of the child in a form. Parents participated by discussing and evaluating video recordings of selected IOS sessions with the art therapist. All participants signed informed consent forms.

### 2.2. Measurement Instruments

In this study, two instruments were used to monitor and describe the children’s and the therapists’ behaviors during the IOS program [26]. (In the evaluative multiple case study three other instruments were also applied: the Behavior Rating Inventory of Executive Functioning (BRIEF; [27,28]), the Children’s Social Behavior Questionnaire (CSBQ) [20,29], and the Self-Perception Profile for Children (SPPC) [21,30]. With the first two instruments, parents’ and teachers’ findings were measured, and with the third instrument, the child’s findings were mapped). The first instrument in the recent study is the so-called ‘Observation in Art Therapy with a child diagnosed with ASD’ (abbreviated as OAT. (In our former studies, the names of the instruments were OAT-A and EAT-A, with ‘-A’ referring to ASD. However, the titles without the suffix ‘-A’ are more compact). The OAT is an instrument that is intended to observe and measure the behavior of the child during art making in the sessions on four subscales: ‘sense of self’, ‘emotion regulation’, ‘flexibility’, and ‘social behavior’.

‘Sense of self’ refers to a theory-based continuum of concepts that represents a development of ‘self skills’: self-perception, self-image, self-concept, and self-esteem [21,31,32]. Improvement of ‘sense of self’ is one of the main aims of AT for children diagnosed with ASD [15,18,33]. In studies about AT for children with ASD the importance of development of the lowest level in the continuum, ‘sense of self’ is often mentioned as a treatment goal [10,11,12]. An example of an item in the OAT subscale ‘sense of self’ is: “*The child is connected with his/her experiences during art making*”.

‘Emotion regulation’ is a complex concept that concerns perception of internal and external stimuli within a complexity of mechanisms: physiological arousal, motivation, and cognitive evaluation. The ability of regulating emotions is based on recognition of inner sensations, feelings, and behavior, relating these to their causes [34]. Initiating, maintaining, inhibiting, or moderating emotional reactions may lead to ‘emotion regulation’ in association with processes that influence experience and the expression of emotions [35,36,37]. ‘Emotion regulation’ refers to the downregulation of negative affects or the upregulation of positive affects [37]. For children with ASD, it is often hard to recognize their own emotions. In art work, inner feelings can be expressed by, for example, drawing cruel fights or monsters. Difficulties with ‘emotion regulation’ can behaviorally be expressed, for instance, by becoming angry when something disappointing happens during art making, or by lack of emotion expression when it is expected. An example of an OAT item of this subscale is: “*The child is expressing emotions/experiences in art making/symbols*”.

‘Flexibility’ is about adaptation to unexpected situations. Cognitive ‘flexibility’ (the ability to find new solutions to a problem) is distinguished from flexible behavior (adaptative skills to unexpected situations) [38]. An example of an OAT item of this subscale is: “*The child uses varied art materials and/or techniques*”.

‘Social behavior’ is one of the main difficulties for children diagnosed with ASD [4]. These children are often hardly or not skilled in reciprocity, to adapt to others, to adapt to play, and in working together [6,29]. In art therapy, some social skills can be developed, for instance task-oriented collaboration, asking for help, and connecting words to experiences [24]. The child may enjoy working together, and joint attention skills may also be developed during art making [39,40,41]. An example of an item in the OAT subscale ‘social behavior’ is: “*The child shows enjoyment during art making with the art therapist*”.

The second instrument is the so-called ‘Evaluation of the Art Therapist’s behavior working with a child diagnosed with ASD’ (abbreviated as EAT); an instrument that is intended to observe and measure art therapeutic behavior in IOS sessions. This instrument has subscales corresponding with those in the OAT: ‘supporting the development of sense of self’, ‘stimulating emotion regulation’, ‘supporting the improvement of flexibility’, and ‘supporting social behavior’. A (corresponding) example of the EAT subscale ‘supporting social behavior’ is: *“The art therapist supports the child to follow directions of the therapist”.*

Items of both instruments have a 5-point Likert scale with values 1 = not observable, 2 = a bit observable, 3 = to some extent observable, 4 = well observable, 5 = strongly observable. Both instruments are filled out by the therapists (see further below). The reliability of the OAT and EAT is ‘moderate’ to ‘substantial’ [26]. Inter-rater reliability of both instruments were determined with pairs of raters (trained art therapists and Bachelor students) who scored selected video fragments of AT sessions collected during a pilot study. Interrater reliability has been computed per item and per subscale of the instruments. Because of the ordinal level of the scores (5-point Likert scale) the degree of agreement was computed per item using quadratic weighted Kappas (K_w_). K_w_ may be influenced by a restriction of the range of scores, resulting in an inflated high or low value. For that reason also Gower indices (G) were computed to interpret values of K_w_ for items with very low or high absolute agreement [42]. In addition: 0.40 ≤ K_w_ < 0.60 means ‘moderate’ agreement; 0.60 ≤ K_w_ ≤ 0.80 means ‘substantial’ agreement.) Training of raters was still proved to enhance the inter-rater agreement regarding the instrument’s scales [26].

### 2.3. Procedure

To include art therapists, a *convenience sample* [41] was drawn using newsletters from the national professional organization of art therapists, calls on Facebook, and word of mouth advertisement. Participating children followed the usual referral procedure from the seven collaborating institutions. Based on the professional judgements of the art therapists, children were excluded if they were assessed as having levels of resistance that were too high or fear of art making.

Both instruments, OAT and EAT, were scored by the art therapist at all of the measurement moments. The use of both instruments was intensively trained and supervised by the PI during the IOS program.

Details regarding the monitoring of treatment were described in our previous publication about IOS evaluation [17].

### 2.4. Analysis

The data were analyzed in two steps. In the first step, the behaviors of each child during the session at the measurement moments T1, T2 and T3 were visually compared with the acts of the corresponding therapists (EAT). The aim was to explore whether changes in the child’s behaviors corresponded to the therapist’s actions. Excel (version 2016 for Windows) was used to analyze the data. As a decision rule, we considered differences between two measurement moments equal to or larger than −1 or +1 as substantial. Differences between −1 and −0.5 or +0.5 and +1 were seen as minor or moderate. Differences smaller than −0.5 or +0.5 were defined as negligible.

In the second step, for each subscale, a nonparametric Friedman test was performed on the four measurement moments for the group of 12 children. The aim of this test was to explore whether there was a consistent pattern of change over time. This enabled comparison of the scores at different measurement moments and visual exploration of the development in children’s and therapists’ behaviors.

## 3. Results

### 3.1. Individual Analyses

Detailed information about changes in child behaviors is shown in Figures 1, 3, and 5; in Figures 2, 4, and 6, the corresponding information is depicted regarding therapist behaviors.

Figure 1 shows the development of the individual children at session 8, halfway through the treatment (T2), compared to T1. Substantial positive developments (≥+1) are identified in one or more subscales for eight children (cases 1, 2, 3, 5, 6, 7, 8, 9), and substantial negative developments (≤−1) in one or more subscales for two children (cases 4, 7). Four children (cases 2, 5, 6, 9) developed on at least two subscales substantially in a positive direction.

In Figure 2, we see four therapists (cases 1, 5, 7, 11) being most active during the time period T1–T2 on the dimension ‘emotion regulation’. Two therapists were substantially active on ‘flexibility’ (cases 5, 12), and one on ‘social behavior’ (case 9).

Figure 3 compares the OAT subscales’ means between the end (T3) and the start (T1) of treatment. Inspecting all four subscales, we see substantial positive change in one or more subscales for ten children (cases 2, 3, 4, 5, 6, 7, 8, 9, 11, and 12) and substantial negative developments in one or more subscales for one child (case 7). Five children (cases 2, 5, 6, 8, and 9) developed substantially on at least two subscales in a positive direction.

The most substantial positive change can be observed on the dimension ‘emotion regulation’ (cases 2, 4, 5, 6, 7, 8, 9, 11, and 12). For the other three subscales, we ascertain substantial positive change in ‘sense of self’, ‘social behavior’ and ‘flexibility’ in four (cases 5, 6, 8, and 9), three (cases 2, 8, and 9), and again three children (cases 2, 3, and 8), respectively. For two children (cases 1 and 10), we hardly observe change, while also observing some negative tendencies.

Figure 4 and Figure 5 show that the art therapists are most actively supportive on ‘social behavior’. For the time period T1–T3, this concerns cases 8 and 9. For T1–T4, this concerns cases 1, 5, 8, 9 and 11. Additionally, for the same time period T1–T4, the art therapist was substantially active in two cases (cases 11 and 12) with respect to ‘stimulating emotion regulation’.

The IOS results 15 weeks after termination of the treatment (T4) compared to T1 are graphically displayed in Figure 6. For ten children (cases 1, 2, 5, 6, 7, 8, 9, 10, 11, and 12), substantial or minor positive change is shown in one or more subscales. Substantial positive change in one or more subscales is shown for six children (cases 1, 2, 5, 7, 8, and 9); substantial negative change in one or more subscales is shown for two children (cases 4 and 7). Looking at all subscales, we ascertain substantial positive change in ‘emotion regulation’ at T4 compared to T1 in five children (cases 1, 2, 5, 7, and 9). For ‘sense of self’, we see substantial positive change in three children (cases 5, 8, 9). Regarding ‘social behavior’ and ‘flexibility’, substantial positive development can be identified in two children (cases 1 and 8). Child 4 showed a substantial negative development in ‘emotion regulation’ and ‘flexibility’. In addition, one child (case 7) developed substantially positive results in ‘emotion regulation’, and substantially negative results in ‘flexibility’.

### 3.2. Change in Children’s Behavior with Respect to Therapists’ Behavior

One purpose of our analysis was to explore whether childrens’ behavior changed during the IOS program, and if and how this was related to the behavior of the therapist. We take a look at this question, thereby focusing on three groups of children: children who seemed to take the most (four cases) or least (one case) advantage of the treatment, and a group that did not exhibit much change (three cases).

*Most profit.* Substantial enduring positive change in more than one subscale is shown in Figure 5 (time period T1–T4) for four children (cases 1, 5, 8, 9). These changes appeared in varied (combinations of) behavior areas. Related to therapist’s behavior mean scores, we saw in these cases that the therapist showed most actions in the ‘supporting social behavior’ area and hardly offered support on developing a ‘sense of self’. Additionally, *during* the therapy sessions (T1–T3), most support from the therapists was directed at the ‘social behavior’ of these children.

*Least profit.* Child 4 showed relatively flat and substantial negative scores for both the time period T1–T2 (Figure 1) and the time period T1–T4 (Figure 5). The profiles (Figure 2, Figure 4 and Figure 6) also show a therapist who is relatively inactive in supporting the child.

*Not much change.* For one dyad (case 10), the profile is quite flat in all compared measurement moments; both the child’s and therapist’s behaviors hardly show any change. Another child (case 3) shows some change on one dimension, ‘flexibility’, during therapy (Figure 1 and Figure 3), but the change is not persistent (Figure 5). In this case, the support of the therapist is at a low level of activity (Figure 2), or even substantially diminished (Figure 4). Child 12 shows some change during the time period T1–T3 in ‘emotion regulation’ (Figure 3); but, looking at the other comparisons of time moments, it is difficult to observe change. The art therapist is surprisingly active in supporting ‘emotion regulation’ at T4 compared to T1 (Figure 6), but this is not associated with positive scores for the child with respect to this behavior (Figure 5).

### 3.3. Group Analyses

The mean ranks of the OAT and EAT of the four measurement moments are shown in Figure 7 and Figure 8. These two figures also make it possible to perform a visual inspection of the change in child behaviors with reference to therapist behaviors as a group.

With respect to the development of the children, in Figure 7, it can be observed that the subscales ‘sense of self’, ‘emotion regulation’ and ‘social behavior’ seem to have a linear development. ‘Sense of self’ (χ^2^(3) = 9.72; *p* = 0.02) and ‘social behavior’ (χ^2^(3) = 13.72; *p* = 0.004) increase significantly, while the linear development in ‘emotion regulation’ is visually apparent, but statistically not significant (χ^2^(3) = 4.70; *p* = 0.20). ‘Flexibility’ shows a constant line (χ^2^(3) = 4.32; *p* = 0.23).

As to the behavior of the therapists, in Figure 8, it can be observed that the subscale ‘stimulating emotion regulation’ seems to have a linear development (χ^2^(3) = 13.50; *p* = 0.004). ‘Supporting sense of self’ also has a significant different pattern from the situation that all measurements moments are the same (χ^2^(3) = 8.52; *p* = 0.04). In this pattern, we observe that after a dip at T2, the subscale increased substantially at T3, but decreased again somewhat at T4. However, the mean rank at T4 is still higher than at T1. The development of ‘stimulating social behavior’ is partly linear; from T1 to T2, this behavior among therapists received more emphasis, which remained at T3 and T4 (χ^2^(3) = 2.64; *p* = 0.45). However, this pattern was not significant. ‘Supporting flexibility’ also shows a constant line in this figure (χ^2^(3) = 2.21; *p* = 0.53).

### 3.4. Patterns in Mean Ranks

The gradients for all four subscales—‘sense of self’, ‘emotion regulation’, ‘flexibility’, and ‘social behavior’—show an upward development between T1 and T2 for the children as well as the therapists, except for one of the therapists’ dimensions: ‘supporting sense of self’ (Figure 8). This indicates that halfway through the treatment, it may be expected that an ‘average child’ would show improvement in all subscales.

Improvement of the children’s behaviors continued at T3 (end of treatment) and even at T4 (15 weeks after ending treatment), with the exception of the dimension ‘flexibility’.

The gradients of the EAT scores for ‘supporting sense of self’ show a decrease at T2, an upward development at T3, and again a decrease at T4. Therefore, especially during the first half of the treatment sessions, the art therapists do not seem to be very active with respect to the ‘sense of self’ dimension.

When comparing the OAT and EAT scores at T2, a comparable pattern can be observed: an increase in mean scores for (supporting) ‘sense of self’, ‘emotion regulation’, ‘flexibility’, and ‘social behavior’. In particular, the heightened EAT scores at T2 are remarkable; the therapists clearly seem to increase their support efforts halfway treatment on ‘emotion regulation’, ‘flexibility’, and ‘social behavior’.

After treatment, only the EAT scores for ‘supporting sense of self’ decreased.

Regarding ‘emotion regulation’, the art therapists’ supportive behavior appeared to be more strongly represented compared to the other three dimensions.

At the end of the treatment, the therapists’ scores seemed to be higher than at the start for the subscales ‘stimulating emotion regulation’ and ‘supporting social behavior’.

## 4. Discussion

In this study, the changes in behavior of children during the IOS program were explored in four dimensions, i.e., ‘sense of self’, ‘emotion regulation’, ‘flexibility’, and ‘social behavior’. Art therapists evaluated the behavior of their young clients and their own behavior by repeatedly completing the OAT and EAT, respectively. Besides descriptive results using each instrument, we also visually explored whether and how changes in the behavior of children were associated with the behavior of therapists.

Looking at the individual children, the change in behavior represented by the subscales ‘sense of self’, ‘emotion regulation’, ‘flexibility’, and ‘social behavior’ was highly varied. It is plausible to firstly explain this variation on the basis of the diversity of problems in children diagnosed with ASD [7,43]. Despite this variation, we see that ten of the twelve children showed moderate to substantial positive behavior changes during art making considering all OAT subscales—at the end of the program and fifteen weeks after treatment termination. This confirms our expectation that the ‘art therapy triangle’ may offer an important contribution; the triangular relationship between the child, the art making, and the therapist seems to give opportunities to improve verbal and nonverbal communication skills [10]. In general, it has been shown that AT is a promising treatment for patients/clients who have difficulties in identifying and expressing their emotions verbally [44]. Interactions with a therapist via artistic means are indicated as being supportive for children with autism-related problems in order to improve their social communication skills [5,27].

Qualitative comments by art therapists can be helpful in better understanding cases with negative changes and those with quite a flat profile (cases 4 and 10, respectively). The art therapists reported that both children improved on most outcomes, but at the same time, they were young persons who remained dependent on a supportive environment. This is a realistic prospect for children diagnosed with ASD [45]. With respect to the child with the flat profile (case 10), the art therapist reported that she observed a development in the art making. Nevertheless, the child kept on saying: “Please explain to me what is the point about this art making?”. This result may indicate that not every child with ASD may develop skills and positive behaviors during art making.

Additionally, some other patterns emerged. For the OAT subscales, the highest amount of substantial and moderate positive change in behavior was established for ‘emotion regulation’: between T1 and T3 for nine children, and between T1 and T4 also for nine children, with an overlap of seven children. This result corresponds with improvement of ‘emotion regulation’ in 50–55% of AT treatment cases, which is in line with a recently published review [5].

In the mean ranks of scores, it is possible to observe an improvement in children’s behaviors, even 15 weeks after treatment, albeit with the exception of the ‘flexibility’ subscale. We notice here that the results regarding children’s behavior during art making differ in some respects from the observations supplied by parents and teachers at home and in the classroom; see also [19]. It may be assumed that the art therapy triangle offers other opportunities for the children’s behavior and expressions during art making and for interacting with the art therapist via the art making [24,46]. Another explanation may be found in the different perspectives and situations of observation of therapists compared to teachers and parents [47].

For the ‘social behavior’ subscale, the moderate or substantial development of eight cases, 15 weeks after treatment, was compared to that of seven cases directly after treatment. The improvement 15 weeks after treatment may indicate an ‘after effect’. Improvement of ‘social behavior’ in 75% of AT treatment cases is in line with a recently published review [5]. For children with ASD, improvement of ‘social behavior’ is often described as one of the most important AT aims [11,12,48].

The EAT ‘supporting sense of self’ scores showed a decrease at T2 compared to T1, an increase at T3, and again a decrease at T4. This may mean that the therapists—in line with the literature—adhere at different moments to varying intensities of actions while offering the child opportunities to learn from tactile experiences. In the literature, it has been found that art therapists presume sensory experiences to be the most supportive element for the child in developing a ‘sense of self’ [10,11,12,15,16]. Improvement of self-esteem, which is part of the ‘sense of self’ continuum, seems to be observed in 50–55% of the AT treatments to be observed [5].

Comparison of mean ranks of observed children’s behaviors with mean ranks of therapists’ behaviors hardly shows a clear relationship between the two behaviors. For instance, we saw little change in efforts in ‘supporting sense of self’ by therapists. An explanation might be that, according to the therapist’s view, the handling and touching of art materials is the main source contributing to improvement of ‘sense of self’ for these children. In other words, and considering the ‘art therapy triangle’ and the communication difficulties of children with ASD, it might be assumed that psychological processes involved in the ‘sense of self’ are not directly influenced by the therapist’s behavior.

The mapping of the first and second IOS phases showed a remarkable positive change in children’s behaviors halfway through treatment. This might indicate that during the first eight sessions, one could already expect positive developments in some or all behavioral dimensions. Additionally, we found an increased activity of the therapists’ halfway treatment on the dimensions of stimulation or supporting ‘emotion regulation’, ‘flexibility’, and ‘social behavior’. It seems plausible that the therapist does not start with full effort, because at the start, it takes time to get acquainted, build a safe situation, etc. When the therapeutic relationship is more or less set, the therapist may increase efforts at T2 and invite the child to share new experiences. At T3, the therapists become again a bit less active in their support, thereby anticipating the moment at which the child needs to further develop without being supported by a therapist and should integrate his/her new experiences, skills and behaviors into daily life. This pattern reminds us of a model of promoting change in people’s behavior, already conceptualized in 1947 by the psychologist Dr. Kurt Lewin in a three-step frame with the phases ‘unfreezing’, ‘moving’, and ‘freezing’. The steps refer to helping someone to orient him/herself to new behavior, to (tentatively) practice it, and to stabilize it with diminishing external support, respectively [49]. Therapists might apply the model even without being professionally aware of its existence.

In their qualitative reported comments, art therapists mentioned that the clearly designed treatment program was helpful. Feedback informed treatment is supportive for therapist’s working in a child psychiatric setting [50]. This contributes to improved quality of life in children with ASD and also supports parents’ expectations. Participating therapists mentioned being surprised that they did not see further change after 15 sessions, because most of them were used to applying art therapy for a period of around a year.

Evaluation moments combined with the use of videos is understood to be supportive for helpers working with ASD patients/clients [51]. Additionally, parents and others who have to communicate with ASD children (teachers) feel supported by the opportunities that video-recordings offers [52].

Our study confirms that working in the ‘art therapy triangle’ allows children with communication problems to develop their sense of self, emotion regulation skills and communication skills, and sometimes also flexibility.

### 4.1. Strengths and Limitations

This study provides new insights into the treatment process by monitoring children’s and therapists’ behavior with respect to the 12 cases that were part of the research. The ‘Images of Self’ program is—as far as we know—the first empirically studied art therapy program, specifically for children diagnosed with ASD. The intervention, with its ‘built-in’ monitoring system, creates opportunities to explore characteristics of the processes of change in the ‘art therapeutic triangle’ of child, art (making), and therapist. The program provides a manual that allows attunement to the client’s needs during art making, in order to build a strong therapeutic alliance via art making.

The combination of the IOS program with the measurement instruments OAT and EAT provided a focus in the treatment for the participating art therapists. The instruments that were applied for observing the children’s and therapists’ behavior were intensively tested on aspects of validity and reliability [26]. The design of the study—a multiple case study with repeated measurements, with a combination of quantitative and qualitative data—allowed us to profile the child’s as well as the therapist’s developments in behavior, and to compare both—case by case and on the level of the whole sample—in our search for associations and patterns.

We also notice some limitations. Although we have confidence in the observation scales OAT and EAT, it cannot be denied that the study mostly leans on the perspective of professionals, i.e., the art therapists. Missing is especially the perspective of the child. While spontaneous utterances of the participating children were mapped, a child-oriented method including ‘their voices’ (like, for instance, by interviewing them) was lacking. This holds less true for the parents and teachers, because they were asked to report their observations at home and at school, respectively.

In addition, there are indications that therapists in some situations might overestimate their competences in therapeutic settings [53]. Contrary to this kind of ‘bias’, we found that participating therapists did not always report their behavior to be as active as might be expected in a therapeutic context. Especially regarding the dimension ‘supporting sense of self’, the level of input by the therapists was relatively low. This did not seem to be evaluated by them as a ‘failure’ of engagement and might argue against bias. Nevertheless, it would be profitable in further research to include neutral, trained observers in order to fill out the OAT and EAT forms—in addition to the observations by the therapists.

The design prohibits the possibility of making causal inferences. If we would like to deepen our insights regarding the question as to what in the therapeutic processes ‘causes’ progress with children, another type of design, i.e., a (quasi-)experimental one, would be needed [54]. This means that a sample of children receiving AT according to the IOS program would be compared to a sample that does were receiving AT, or that were receiving another treatment. Considering the necessary ‘power’ to make valid inferences, a bigger sample than the current one is required.

As indicated up here we found some differences between the results of this study compared to our former study [18]. This may indicate that the observed behaviors of the children during art making are different from the behaviors at home and in the classroom. For that reason we suggest to include all relevant contexts (therapy, home, school, leisure time) in future research.

### 4.2. Recommendations

We recommend continuing the research on art therapy by applying the IOS program on a larger scale and carefully monitoring the results. Above, we already argued in favor of a (quasi-)experimental design to gain further insight into the effectiveness of IOS. More detailed insights into children’s and therapists’ behavior during AT and how these relate to each other can be gained from a larger pool of *n* = 1 studies with repeated measurements. In these studies, the ‘voice of the child’ should be included—more than was the case in the current research.

Special attention should be paid to the working mechanisms of art making as an instrument to improve specific behaviors: what exactly is the role of making art and expressing oneself in an artistic way for the child’s development and behavior? A combination of narrative methods (interview, diary, focus group) and content analysis of ‘art products’ might be helpful in further clarifying the dynamics during AT.

In a more practical sense, we propose to improve the IOS program by gathering feedback from experienced therapists and trainees, as well as from parents or other network members like teachers. As was already indicated, the use of video-recordings, together with analyzing and discussing these afterwards, has proven to be very valuable. For that reason, we consider the instruments OAT and EAT that have been used in this study to systematically observe children’s and therapists’ behavior, as an integral part of the IOS program. The implication is that AT therapists using IOS should be thoroughly trained in the implementation of these instruments.

Until now, the IOS program has only been applied and studied in the context of mental health care services for children with ASD. Recently, a pilot study started to broaden the field in which IOS could be applied. An empirical study to investigate the possibilities and opportunities to apply the program in a school context, thereby studying the preventive qualities of AT for children with ASD and other psychosocial problems: is participation of vulnerable children in IOS during school hours helpful in preventing their referral to more ‘heavy’ psychosocial services or treatments? In addition, in what way does IOS, applied in an educational environment for teachers, support expanded possibilities for accompanying children with ASD [9]. 

## Figures and Tables

**Figure 1 children-09-01036-f001:**
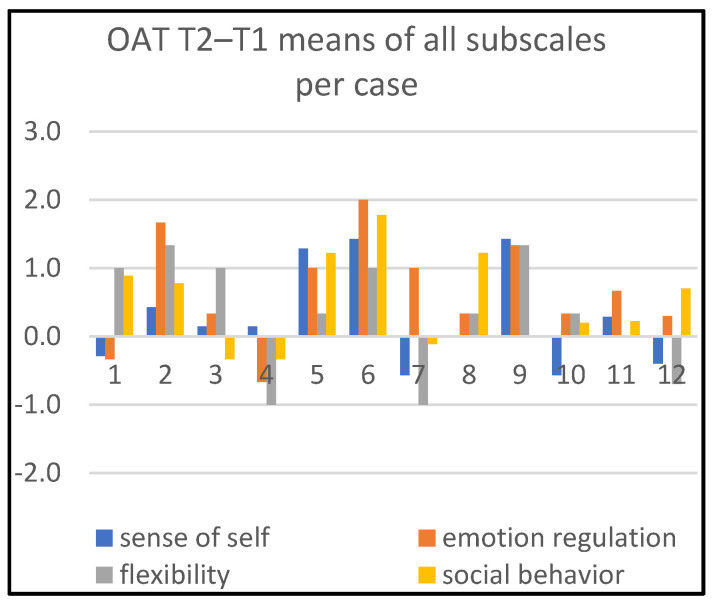
Means of all subscales per case, T2–T1.

**Figure 2 children-09-01036-f002:**
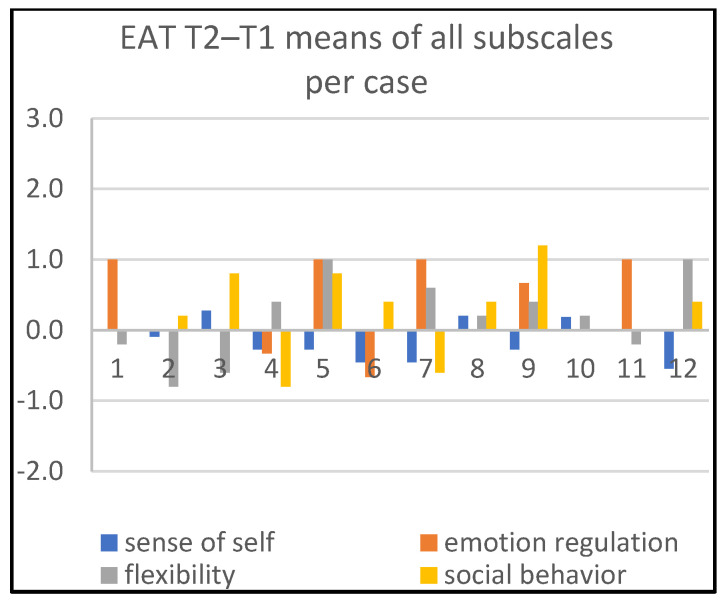
Means of all subscales per case, T2–T1.

**Figure 3 children-09-01036-f003:**
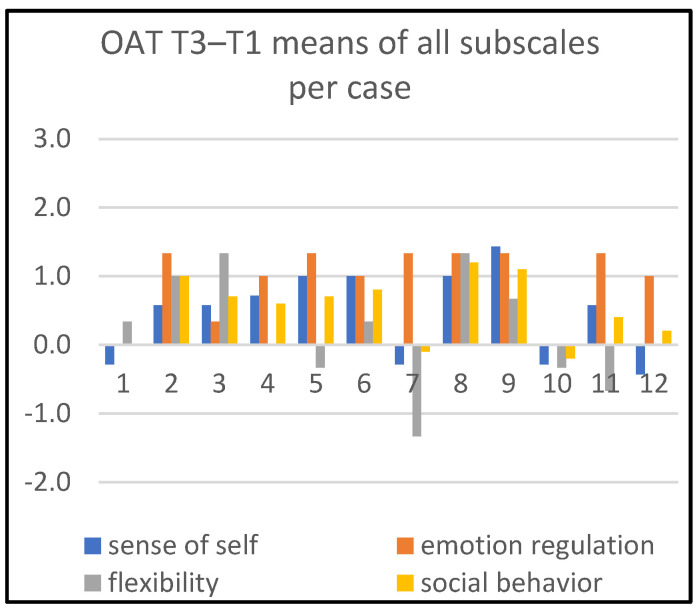
Means of all subscales per case, T3–T1.

**Figure 4 children-09-01036-f004:**
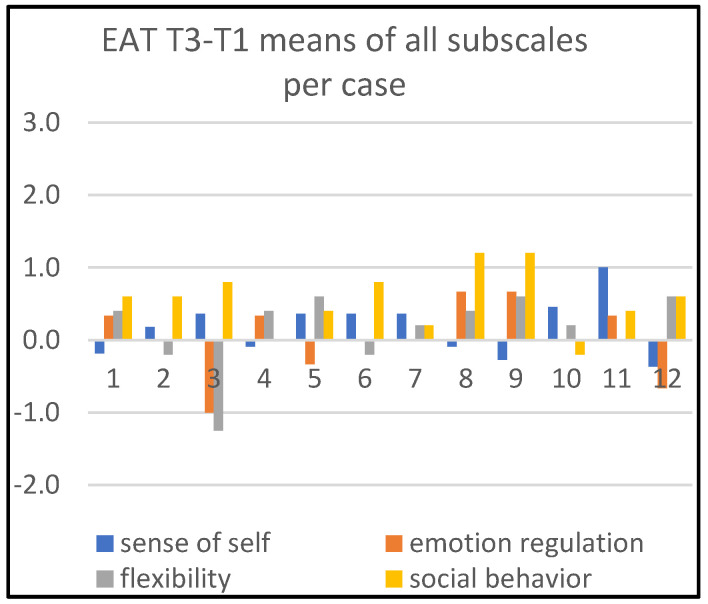
Means of all subscales per case, T3–T1.

**Figure 5 children-09-01036-f005:**
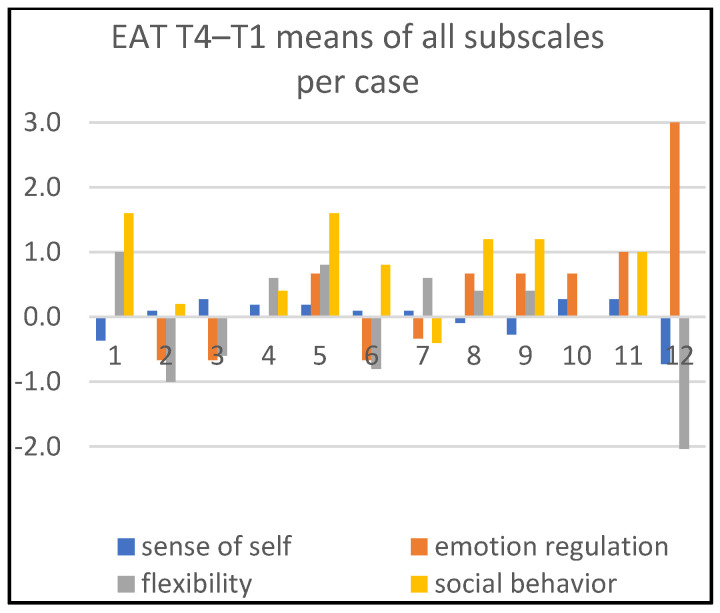
Means of all subscales per case, T4–T1.

**Figure 6 children-09-01036-f006:**
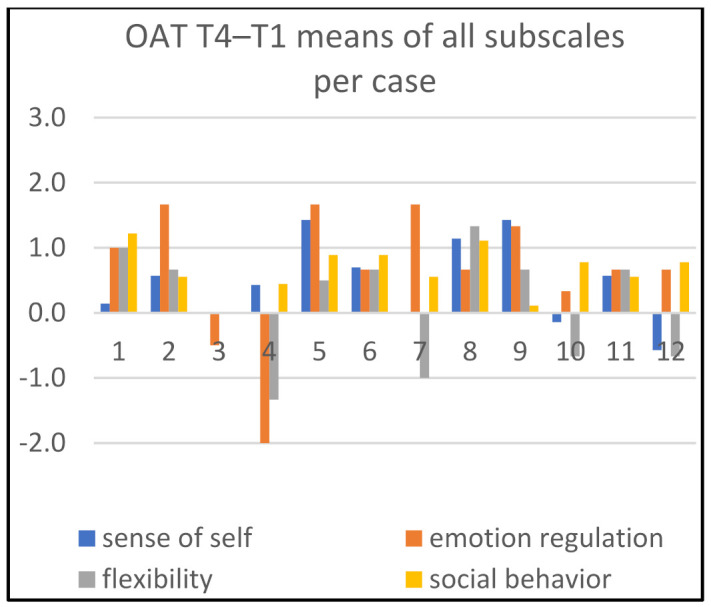
Means of all subscales per case, T4–T1.

**Figure 7 children-09-01036-f007:**
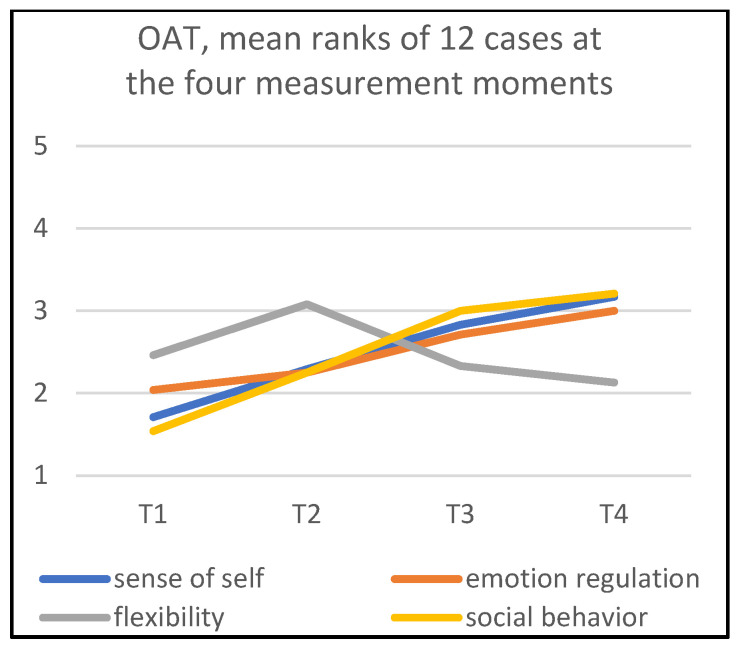
Mean ranks of OAT.

**Figure 8 children-09-01036-f008:**
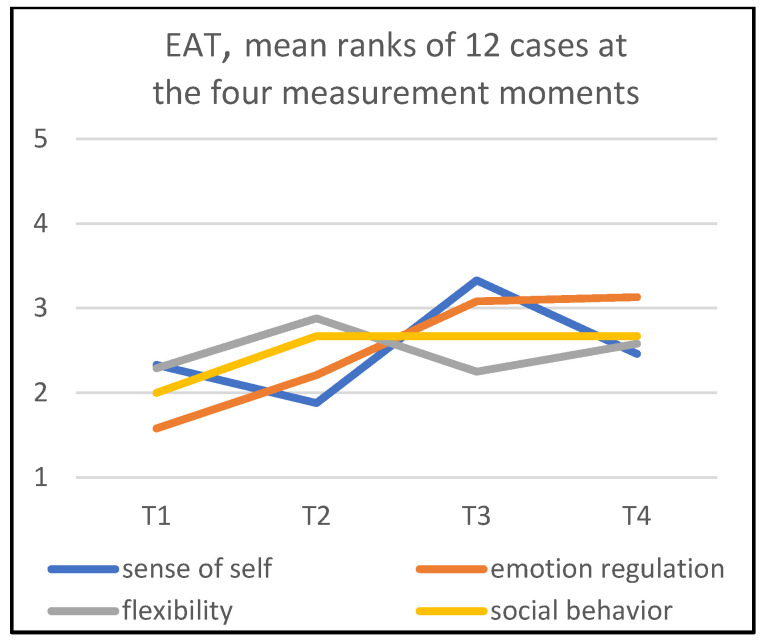
Mean ranks of EAT.

## Data Availability

The data presented in this study are available on request from the corresponding author.

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
