# Peer review of "Exploring Change in Children’s and Art Therapists’ Behavior during ‘Images of Self’, an Art Therapy Program for Children Diagnosed with Autism Spectrum Disorders: A Repeated Case Study Design"

_children, 2022, doi:10.3390/children9071036_

Round 1
Reviewer 1 Report
The study is very interesting and certainly a valuable addition to the field.
It is hard to study children with autism and draw generalisation; therefore it would be helpful to have the word 'case-studies' emphasised at the start of the article (perhaps even in the title). The sample if relatively small and it is hard to draw generalisations from it so this would help to clarify it.
In the Introduction, highlighting other studies on art therapy and autism would make the paper even stronger. In the Results section, some other graphs in addition to the current ones might help the reader to understand the findings. For example, simple charts for deviation, mean and mode might help.
Otherwise, the paper reads very well and is very interesting.
Author Response
To: Reviewer 1 for the article, titled: Exploring change in children and art therapist’s behavior during ‘Images of Self’, an art therapy program for children diagnosed with Autism Spectrum Disorders. Submitted for the Special Issue ‘Art Therapy’ in ‘Children’
Subject: Response to reviewer 1.
From: celine.schweizer@nhlstenden.com
Phone: 0031-6-1928-1378
Leeuwarden, Netherlands,
June 30 – 2022
Dear Reviewer 1,
Hereby we would like to thank you for your comments to our article, titled: Exploring change in children and art therapist’s behavior during ‘Images of Self’, an art therapy program for children diagnosed with Autism Spectrum Disorders.
We have adapted the article following your comments:
- The study is very interesting and certainly a valuable addition to the field. Thank you for the compliments!
- It is hard to study children with autism and draw generalisation; therefore it would be helpful to have the word 'case-studies' emphasised at the start of the article (perhaps even in the title). The sample if relatively small and it is hard to draw generalisations from it so this would help to clarify it.
Thank you for this valuable suggestion! We added a line in the title about casestudies. Also in the text we added information in the following lines of the manuscript: line 14 (abstract; 109-112 (method)
- In the Introduction, highlighting other studies on art therapy and autism would make the paper even stronger.
We described more about the content from the studies that were already mentioned in the text (lines 36-47). .
- In the Results section, some other graphs in addition to the current ones might help the reader to understand the findings. For example, simple charts for deviation, mean and mode might help.
We understand that it may be helpful to have a better understanding about the Figures. We realize that it may take some time to understand the Figures. We have tried several versions to provide the most clear information in the Figures and chose for Figures that show differences between means of the outcomes on the measure moments. Unfortunately it is impossible to calculate SD from differences between measurement moments.
- Otherwise, the paper reads very well and is very interesting. Thank you!
We hope the changes are according to your expectations.
Yours sincerely, also on behalf of the co-authors,
Erik J. Knorth, PhD,
Tom. A. Van Yperen, PhD,
Marinus Spreen, PhD.
Celine Schweizer, MA, Corresponding author.

Reviewer 2 Report
Thank you very much for the opportunity to read and evaluate this article. The author(s) do not in great detail describes the possibilities of support and development of self-image in children and ASD when using art therapy techniques in practice. The topic of the article is very interesting for two professional areas: Special education and Art therapy. The author (s) presented the results of the research they carried out. The presentation of the results is clear. The statistics data are supplemented by qualitative analysis. Though the article has the quite good quality I have requirement
of two minor revisions:
1. Please, could you improve the Introduction part the supplement the theoretical background of your topic and used more different authors for characteristics of the used terms.
2. Please, could you show an example of used Art therapy practice. The readers will be interested to know what precisely helps to improve Images of Self in children with ASD.
Author Response
To: Reviewer 2 for the article, titled: Exploring change in children and art therapist’s behavior during ‘Images of Self’, an art therapy program for children diagnosed with Autism Spectrum Disorders. Submitted for the Special Issue ‘Art Therapy’ in ‘Children’
Subject: Response to reviewer 1.
From: celine.schweizer@nhlstenden.com
Phone: 0031-6-1928-1378
Leeuwarden, Netherlands,
June 30 – 2022
Dear Reviewer 2,
Hereby we would like to thank you for your comments to our article, titled: Exploring change in children and art therapist’s behavior during ‘Images of Self’, an art therapy program for children diagnosed with Autism Spectrum Disorders.
We think the comments were useful and hereby we adapted the article following your comments:
- Thank you very much for the opportunity to read and evaluate this article.
Thank you for your efforts and valuable comments.
- The author(s) do not in great detail describes the possibilities of support and development of self-image in children and ASD when using art therapy techniques in practice.
We added an explanation about the title ‘Images of Self’. The title may suggest that the program concerns self-image, but the concept refers to a broader conceot (see the added lines 51-54, and also see lines 120 and further)
- The topic of the article is very interesting for two professional areas: Special education and Art therapy. The author (s) presented the results of the research they carried out. The presentation of the results is clear. The statistics data are supplemented by qualitative analysis. Thank you!
- Though the article has the quite good quality I have requirement
of two minor revisions:
Please, could you improve the Introduction part the supplement the theoretical background of your topic and used more different authors for characteristics of the used terms. Thank you for the suggestion. We added some authors that underline different concepts of autism related problems (references 54, 55 and 56).
2. Please, could you show an example of used Art therapy practice. The readers will be interested to know what precisely helps to improve Images of Self in children with ASD.
Thank you for this valuable suggestion. We added an example from our qualitative data (lines 63-73).
We hope the changes are according to your expectations.
Yours sincerely, also on behalf of the co-authors,
Erik J. Knorth, PhD,
Tom. A. Van Yperen, PhD,
Marinus Spreen, PhD.
Celine Schweizer, MA, Corresponding author.
